# Revisiting Dynamic Graphs from the Perspective of Time Series

## Abstract

Numerous studies have been conducted to investigate the temporal pattern of dynamic graphs. Existing methods predominantly fall into two categories: discrete-time dynamic graph (DTDG) methods and continuous-time dynamic graph (CTDG) methods. While these approaches have proven effective in modeling temporal dependencies within dynamic graphs, they exhibit several limitations. For instance, DTDG approaches often lose fine-grained temporal information. CTDG methods can preserve temporal details but may inadequately capture long-term dependencies due to computational constraints. Moreover, both paradigms predominantly focus on existing historical interactions, often neglecting the informative value of non-existing ones. These negative historical interactions can provide complementary insights into the recurring patterns of node behavior. To fully leverage both types of interactions, we propose transforming node interactions into binary time series. Building upon this formulation, we propose a novel model termed the **T**ime **S**eries-based **D**ynamic **G**raph (**TSDyG**) model, which approaches dynamic graph learning from a time series perspective. Compared to existing DTDG and CTDG methods, our model offers several advantages: it captures long-range dependencies, preserves fine-grained temporal details, and leverages information from both existing and non-existing historical interactions. We conduct extensive evaluations of our method on various benchmark datasets. The results demonstrate that our proposed TSDyG model achieves competitive performance on the downstream task such as link prediction.

## 1 Introduction

Dynamic graphs model evolving systems in which interactions between entities change over time. Many real-world scenarios, such as social networks, user-item interactions, and financial transactions, can be naturally represented as dynamic graphs. In recent years, a growing number of research (Zhang et al., 2024; 2023; Ji et al., 2024; Cong et al., 2023) on dynamic graph learning has emerged, demonstrating its effectiveness in capturing temporal relationships among entities and achieving promising results in forecasting tasks.

Current dynamic graph learning methods can generally be categorized into two types: discrete-time dynamic graph (DTDG) methods (Karmim et al., 2024; You et al., 2022; Yang et al., 2021; Sankar et al., 2020; Pareja et al., 2020) and continuous-time dynamic graph (CTDG) methods (Yu et al., 2023; Tian et al., 2023; Zhang et al., 2024; 2023; Ji et al., 2024; Zou et al., 2024; Poursafaei et al., 2022; Gravina et al., 2024). In DTDG methods, the dynamic graph is represented as a sequence of snapshots that are in the form of static graphs to capture the interactions of entities during the specific time interval. These models typically employ Graph Neural Networks (GNNs) (Kipf & Welling, 2017; Hamilton et al., 2017; Xu

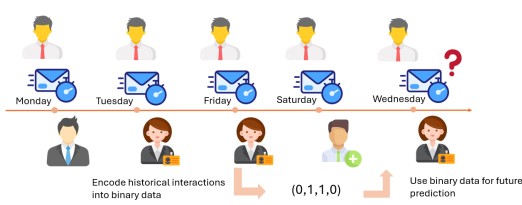

Figure 1: Illustration of encoding historical interactions into binary data for future prediction. For the target entities (the man and the woman), interactions on Tuesday and Friday are encoded as "1," while the absence of interactions at other times is encoded as "0."

et al., 2019) in conjunction with Recurrent Neural Networks (RNNs) (Hochreiter & Schmidhuber, 1997; Cho et al., 2014) to capture both the structural and temporal dependencies in the evolving graph. However, DTDG methods exhibit several limitations (Fennell et al., 2016; Cho et al., 2014). First, due to the partitioning of interactions into discrete snapshots, fine-grained temporal information is lost, which can negatively impact performance in time-sensitive prediction tasks. Second, selecting an appropriate snapshot interval is non-trivial: if the interval is too small, it can lead to redundant and computationally expensive graph sequences; if too large, important temporal details may be overlooked. In addition to these challenges, scalability remains a concern, especially on large-scale dynamic graphs.

Continuous-time dynamic graph (CTDG) methods, in contrast to DTDG approaches, represent dynamic graphs as sequences of chronologically ordered events (Yu et al., 2023; Wang et al., 2021a). Two main categories of CTDG methods have been developed: model-centric and memory-based approaches. Compared to DTDG methods, CTDG models can better preserve fine-grained temporal information. However, CTDG methods also face several limitations. Model-centric approaches (Yu et al., 2023; Zou et al., 2024; Wu et al., 2024), such as those based on Transformers, often struggle to capture long-range temporal dependencies due to their high computational complexity over continuous event streams. On the other hand, memory-based methods (Ji et al., 2024; Su et al., 2024; Rossi et al., 2020), typically exhibit inferior performance because they process batches of events concurrently rather than sequentially,

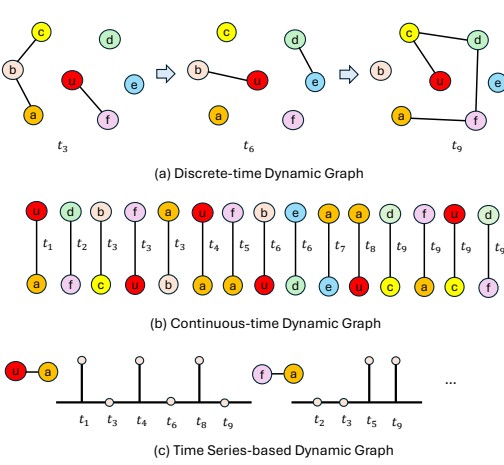

(a) Discrete-time Dynamic Graph

(b) Continuous-time Dynamic Graph

(c) Time Series-based Dynamic Graph

Figure 2: The illustration of discrete-time dynamic graph, continuous-time dynamic graph and our proposed time series-based dynamic graph.

violating the natural chronological order of interactions, a challenge often referred to as temporal discontinuity (Su et al., 2024).

In dynamic graph modeling, both discrete-time and continuous-time approaches primarily capture temporal dependencies by focusing on positive interactions between target nodes and their recent historical neighbors, often overlooking the informative value of non-existing historical interactions. Given a sequence of interactions $\mathcal{G} = \{(u_1, v_1, t_1), (u_2, v_2, t_2), \dots), (u_T, v_T, t_T)\}$ with $0 \le t_1 \le t_2 \le \dots \le t_T$, the absence of interaction at a specific time $t'$, where $(u, v, t') \notin \mathcal{G}$, can be equally informative. Such negative interactions capture the temporal recurring patterns of interactions, reflecting behaviors like periodicity or seasonality. For instance, as shown in Example 1, the illustration depicts email communications between employees. Our goal is to predict whether the target entities (the man and the woman) will exchange emails next Wednesday. In such scenarios, the absence of interaction (e.g., no email communication between the man and the woman on the previous Monday and Saturday) also provides useful information for modeling their behavior. Both existing and non-existing interactions form a predictable pattern that models should capture. To effectively capture both existing and non-existing interactions of target entities over time, we propose transforming the interaction data into binary time series. Given the target node $u$ and a historical node $v$, their past interactions over time can be represented by the proposed function $f_{u,v}(t)$, which captures the interaction dynamics as a function of time and can be defined as:

$$f_{u,v}(t) = \begin{cases} 1, & \text{if } (u, v, t) \text{ or } (v, u, t) \in \mathcal{G} \\ 0. & \text{otherwise} \end{cases} \tag{1}$$

This binary time series $\{f_{u,v}(t)\}_{t=t_1}^{t_T}$ encapsulates the complete interaction history between $u$ and $v$, enabling models to learn from both the presence and absence of interactions. Incorporating negative historical interactions in this manner allows for a more comprehensive understanding of temporal dynamics. The difference of discrete-time dynamic graphs, continuous-time dynamic graphs and our proposed time series-based dynamic graphs are illustrated in Figure 2.

Enlightened by this transformation, we propose a novel dynamic graph learning method named **T**ime **S**eries-based **Dy**namic **G**raph (TSDyG) model which handles the dynamic graph from the perspective of time series. TSDyG comprises three key components: a time series formulation module, an embedding generation module, and a cross-attention module. In the time series formulation module, we convert node interactions into binary time series as defined in Eq. 1. Each time step indicates the presence (1) or absence (0) of an interaction between node pairs. Next, the embedding generation module employs a learnable codebook with two learnable embeddings to generate the interaction embeddings from the binary time series data, To incorporate temporal information, a time encoder is adopted to generate time-specific embeddings, which are combined with the interaction embeddings to form the input for the cross-attention module. The cross-attention module, drawing inspiration from prior work (Kim et al., 2024), introduces a learnable query token that interacts with the key-value pairs derived from the input embeddings. This design facilitates the modeling of long-range temporal dependencies from historical data while maintaining lower computational complexity compared to traditional self-attention mechanisms. During training, our model is optimized with the binary cross-entropy (BCE) loss. Compared to the previous DTDG and CTDG methods, TSDyG is distinguished by its ability to leverage both existing and non-existing interactions to model the recurring interaction patterns among nodes, while effectively capturing long-term dependencies in dynamic graphs. The contributions of our paper are summarized as follows:

- Unlike previous DTDG and CTDG methods that treat dynamic graphs as sequences of snapshots or discrete events, we introduce a novel formulation that represents dynamic graphs as time series. This formulation captures both existing and non-existing historical interactions of target nodes, offering a more comprehensive perspective on node dynamics.

- Building on the formulated binary time series data, we propose the Time Series-based Dynamic Graph (TSDyG) model, which comprises three key components. In contrast to previous dynamic graph methods, TSDyG effectively captures recurring interaction patterns between nodes and models long-term temporal dependencies in dynamic graphs.

- We extensively evaluate our model on multiple benchmark datasets, and the results demonstrate that it achieves competitive performance on downstream tasks, such as link prediction, compared to the baselines.

## 2 RELATED WORK

**Dynamic Graph Learning.** Existing methods can roughly categorized into discrete-time and continuous-time approaches. Discrete-time methods (Karmim et al., 2024; You et al., 2022; Yang et al., 2021; Sankar et al., 2020) regard dynamic graphs as a sequence of snapshots taken at regular time intervals, and typically extend the graph neural networks (GNNs) for static graphs to capture the temporal correlations. Recent work (Karmim et al., 2024) has explored graph transformers as a powerful alternative to GNN for modeling node dependencies. However, discrete-time methods usually suffer some significant limitations, such as the loss of temporal information. In contrast, continuous-time methods (Zhang et al., 2024; Zou et al., 2024; Poursafaei et al., 2022; Gravina et al., 2024) represent dynamic graphs as the chronologically ordered sequences of events. Among the continuous-time methods, memory-based methods (Ji et al., 2024; Su et al., 2024; Rossi et al., 2020) maintain a memory to update the node states based on interactions. However, during batch processing, the strict chronological order of the events may be violated. Model-centric methods (Yu et al., 2023; Zou et al., 2024; Wu et al., 2024) leverage sequential models such as LSTMs (Hochreiter & Schmidhuber, 1997), Transformers (Vaswani et al., 2017) and MLP-Mixers (Tolstikhin et al., 2021)) to capture long-range node dependencies while aiming to reduce the time complexity. Other methods have proposed techniques like temporal walk (Wang et al., 2021b; Jin et al., 2022) and graph ordinary differential equation (graph ODE) (Gravina et al., 2024; Luo et al., 2023) for dynamic graph representation learning. Additionally, several studies (Yuan et al., 2024; Yang et al., 2024) have shown that existing dynamic graph methods often struggle to generalize under distribution shifts, prompting the development of new techniques to address these challenges.

**Time Series Forecasting.** Time series forecasting is one of the fundamental tasks in time series analysis. Traditional statistical approaches, such as VAR (Watson, 1994) and ARIMA (Box et al., 1974) are often inadequate when dealing with non-linear temporal dynamics. In contrast, deep learning methods have demonstrated strong capabilities in capturing complex temporal patterns. Based

on their architectural backbones, these methods can be broadly classified into four categories: CNN-based, RNN-based, Transformer-based, and MLP-based models. CNN-based methods (Liu et al., 2022) utilize convolution kernels to model local temporal variations. However, due to their limited receptive fields, they struggle to capture long-term dependencies. RNN-based methods (Salinas et al., 2020; Lai et al., 2018) model the temporal state Transition via recurrent structure. In comparison, transformer-based methods (Kitaev et al., 2020; Zhou et al., 2021; Kim et al., 2024; Liu et al., 2024; Nie et al., 2023; Zhang & Yan, 2023) achieve superior performance in forecasting tasks by introducing techniques like patching for efficient modeling of long-range dependencies. More recently, inspired by the MLP-based method (Zeng et al., 2023; Wang et al., 2024), recent work (Kim et al., 2024) further demonstrates that cross-attention is more effective than self-attention in time series forecasting. Beyond time-domain approaches, there is also a growing body of work (Zhou et al., 2022; Wang et al., 2025; Eldele et al., 2024; Yi et al., 2023) focusing on frequency-domain modeling, which seeks to capture temporal patterns using spectral techniques. These frequency-aware methods (Zhou et al., 2022; Wang et al., 2025; Eldele et al., 2024) have achieved competitive results and offer a complementary perspective to traditional time-domain forecasting models.

## 3 PRELIMINARY

**Discrete-time Dynamic Graph (DTDG).** The discrete-time dynamic graph is represented as a sequence of snapshots $\mathcal{G} = \{G_1, G_2, \dots\}$, where each snapshot $G_t = (\mathcal{V}_t, \mathcal{E}_t)$ is a static graph sampled at regular time intervals. $\mathcal{V}_t \subseteq \mathcal{V}$ denotes the set of active nodes at timestamp $t$, where $\mathcal{V}$ is the complete node set, and $\mathcal{E}_t \subseteq \mathcal{V} \times \mathcal{V}$ represents the set of observed edges at timestamp $t$.

**Continuous-time Dynamic Graph (CTDG).** The continuous-time dynamic graph usually consists of non-decreasing chronological events $\mathcal{G} = \{(u_1, v_1, t_1), (u_2, v_2, t_2), \dots, (u_T, v_T, t_T)\}$, where $0 \le t_1 \le t_2 \le \dots \le t_T$. Each triplet $(u_i, v_i, t_i)$ signifies an interaction between source node $u_i \in \mathcal{V}$ and destination node $v_i \in \mathcal{V}$ at timestamp $t_i$.

**Time series-based Dynamic Graph (TSG).** We define a time series-based dynamic graph by converting node interactions into binary time series. For each pair node $(u, v) \in \mathcal{V} \times \mathcal{V}$, we define its interaction series as $\{f_{u,v}(t)\}_{t=t_1}^{t_T}$, where $f_{u,v}(t) \in \{0, 1\}$ indicates whether an interaction occurred between node $u$ and $v$ at timestamp $t$. The function $f_{u,v}(t)$ is formally defined in Eq. 1.

For attributed dynamic graphs, each interaction $(u, v, t)$ is associated with an edge feature $e_{u,v}^t \in \mathbb{R}_E^d$, where $d_E$ denotes the dimension of the edge feature. If the graph is non-attributed, the edge feature is simply set to zero vectors.

**Problem Formalization.** Given the formulated time series of the source node $u$ and destination node $v$ prior to timestamp $t$, representation learning on the time series-based dynamic graphs aims to develop a model that learns time-aware representations that capturing the temporal patterns of their interactions. The effectiveness of the learned representation is evaluated through the link prediction.

## 4 METHODOLOGY

In this section, we introduce our proposed TSDyG. TSDyG is composed of three core components: a time series formulation module, an embedding generation module, and a cross-attention module. The overall architecture of TSDyG is illustrated in Figure 3.

**Time Series Formulation Module.** Given the historical interactions of source node $u$ and destination node $v$, the time series formulation module aims to construct the time series leading up to the current timestamp $t_c$. However, selecting appropriate timestamps is a non-trivial task. Naively including all timestamps before $t_c$ is suboptimal for two reasons. First, when $t_c$ is large, the time sequence can become excessively long, making the model difficult to process effectively. Second, for node pairs with sparse interactions, the time series may contain little meaningful information. Conversely, randomly sampling timestamps may omit important interactions information. Therefore, constructing a time series that is both tractable and informative requires a careful design. To address this, we simply adopt the existing temporal neighbor sampling method and select only those timestamps at which an interaction involving either the source or destination node occurs. This design choice is motivated by two key considerations. First, timestamps without any interactions in-

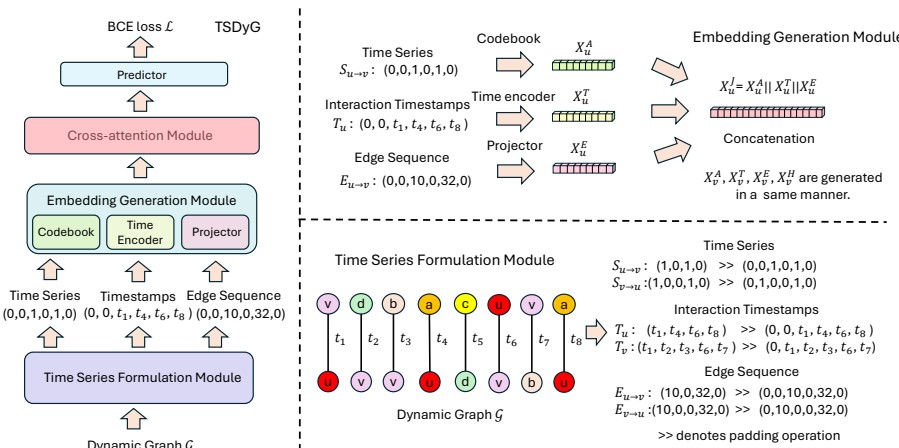

Figure 3: The overview of the proposed Time Series-based Dynamic Graph (TSDyG) model. TS-DyG comprises three main components: (1) the time series formulation module, which generates binary time series, interaction timestamps, and edge sequences from dynamic graphs; (2) the embedding generation module, which encodes these inputs into interaction, time, and edge embeddings; and (3) the cross-attention module, which models temporal evolution by extracting informative patterns from the time series to produce time-aware node representations.

volving the source or destination node provide little to no information about their temporal behavior and thus are irrelevant for modeling interaction patterns. Second, by focusing on timestamps with actual interactions, we can identify the counterpart nodes involved, which helps the model capture nuanced behavioral patterns of the target nodes.

Based on this observation, we sample the timestamps at which actual interactions involving either the source or destination node occur. Specifically, for source node $u$, we define the interaction timestamps as $T_u = \{t | (u, o, t) \text{ or } (o, u, t) \in \mathcal{G}, o \in \mathcal{V}, t < t_c\}$. For efficient batch processing, we retain the most recent $N$ timestamps from $T_u$. Using these timestamps, we construct a binary time series sequence for target node $u$ with respect to neighboring node $v$, denoted as $S_{u \to v} = f_{u,v}(T_u) \subseteq \{0, 1\}^N$, where each entry indicates whether an interaction occurs between nodes $u$ and $v$ at the corresponding timestamp. For example, suppose target node $u$ had historical interactions from $t_1$ to $t_6$. and only interacted with neighboring node $v$ at $t_3$ and $t_5$. Then, the resulting binary time series would be $\{0, 0, 1, 0, 1, 0\}$. If the sequence length is shorter than $N$, zero-padding is applied to maintain a consistent length.

For attributed dynamic graphs, we can also derive the corresponding edge ID sequences for source node $u$ with respect to node $v$, denoted as $E_{u \to v} = f_{u,v}^e(T_u) \subseteq \mathbb{N}^N$. The function $f_{u,v}^e(T_u)$ is defined as:

$$f_{u,v}^e(t) = \begin{cases} e_t, & \text{if } (u, v, t) \text{ or } (v, u, t) \in \mathcal{G} \\ 0, & \text{otherwise} \end{cases} \tag{2}$$

where $e_t$ denote the edge ID at timestamp $t$. Similarly, we can obtain $T_v, S_{v \to u}$ and $E_{v \to u}$ for destination node $v$ in the same manner.

**Embedding Generation Module.** In embedding generation module comprises three components: a discrete codebook with two entries, a time encoder and projection layers. These components are responsible for generating the interaction embedding, time embedding and edge embedding from the binary time series, interaction timestamps and the edge sequences produced by the time series module. The codebook consists of two learnable vectors representing the presence or absence of an interaction between the node pair. And the interaction embedding can be extracted from the codebook by indexing it with entries from $S_{u \to v}$. For instance, a value of 0 corresponds to the first entry in the codebook. For source node $u$, the projected interaction embedding is computed as $X_u^A = \tilde{X}_u^A W_A \in \mathbb{R}^{N \times d_C}$, where $\tilde{X}_u^A = \text{codebook}(S_{u \to v}) \in \mathbb{R}^{N \times d_A}$ is the codebook output corresponding to the binary time series $S_{u \to v}$ and $W_A \in \mathbb{R}^{d_A \times d_C}$ is the weight matrix of the interaction embedding projector. Here, $d_A$ and $d_C$ denote the dimensions of the codebook vector and the projected embedding, respectively. To capture the temporal information of the evolving interaction patterns, we adopt a time embedding proposed by previous work (Cong et al., 2023).

The $i$-th entry of the time embedding for source node $u$ is formulated as:

$$\tilde{X}_u^T[i] = \sqrt{\frac{1}{d_T}}[\cos(w_1\Delta t_i), \cos(w_2\Delta t_i), \ldots, \cos(w_{d_T}\Delta t_i)], \tag{3}$$

where $\Delta t_i = t_c - t_i$ is the time interval between the current timestamp $t_c$ and the $i$-th timestamp $t_i \in T_u$. $[w_1, w_2, \ldots, w_{d_T}]$ are trainable parameters, and $d_T$ denotes the dimension of the time embedding. The projected time embedding is obtained via linear transformation: $X_u^T = \tilde{X}_u^T W_T \in \mathbb{R}^{N \times d_C}$, where $W_T \in \mathbb{R}^{d_T \times d_C}$ represents the weight matrix of the time embedding projector.

The projected edge embedding is computed as: $X_u^E = \tilde{X}_u^E W_E \in \mathbb{R}^{N \times d_C}$, where $\tilde{X}_u^E \in \mathbb{R}^{N \times d_E}$ denotes the raw edge embeddings, and $W_E \in \mathbb{R}^{d_E \times d_C}$ is the weight matrix of the edge embedding projector. $d_E$ represents the dimension of raw edge embeddings. The joint embedding for source node $u$ is constructed by concatenating projected interaction, time and edge embedding. For attributed dynamic graph, $X_u^J = X_u^A || X_u^T || X_u^E \in \mathbb{R}^{N \times 3d_C}$ or $X_u^J = X_u^A || X_u^T \in \mathbb{R}^{N \times 2d_C}$ for non-attributed dynamic graph. We apply a linear transformation on joint embedding to obtain the input embedding $X_u^H = X_u^J W_H \in \mathbb{R}^{N \times d_H}$, where $W_H \in \mathbb{R}^{d_J \times d_H}$ is the projection matrix (with $d_J = 3d_C$ for attributed or $2d_C$ for non-attributed) and $d_h$ denotes the hidden dimension of the subsequent model layers. The corresponding embeddings for destination node $v$, i.e., $X_v^A, X_v^T, X_v^E, X_v^J$ and $X_v^H$ are computed in the same manner.

**Cross-attention Module.** Inspired by the previous work, we introduce the cross-attention mechanism to model the temporal patterns of the time series. Specifically, we use a learnable latent token $Z^L \in \mathbb{R}^{1 \times d_H}$ as the query which interacts with the key and value representations derived from the time series embedding. This design allows the model to distill the most relevant temporal information. Compared to self-attention, cross attention has a linear time complexity $\mathcal{O}(Nd_H^2)$, enabling our model to efficiently capture long-range temporal dependencies. For the source node $u$, the processing pipeline in the cross-attention module is illustrated as follows:

$$\begin{aligned}
Z_u^0 &= Z_u^L, \\
Q_u^{i-1} &= Z_u^{i-1} W_Q, \ K_u^{i-1} = Z_u^H W_K, \ V_u^{i-1} = Z_u^H W_V, \\
Z_u^{i-1} &= \text{cross-attention}(Q_u^{i-1}, K_u^{i-1}, V_u^{i-1}), \\
Z_u^i &= \text{LN}(\text{FFN}(Z_u^{i-1}) + Z_u^{i-1}), \quad (i = 1, 2, 3),
\end{aligned} \tag{4}$$

where LN denotes layer normalization. The final output embedding for source node $u$ is denoted as $Z_u^O = Z_u^3$. The final output embedding for destination node $u$ is obtained using the same process.

**Training.** To predict the likelihood of an interaction between the source node $u$ and the destination node $v$, we employ a multi-layer perceptron (MLP) predictor that takes their final output embeddings as input: $\tilde{p} = \text{MLP}(Z_u^O, Z_v^O)$. The model is trained using the binary cross-entropy loss $\mathcal{L} = -\frac{1}{M}\sum_{i=1}^{M}(p_i \log(\tilde{p}_i) + (1 - p_i)\log(1 - \tilde{p}_i))$, where $M$ denotes the number of training samples (including both positive and negative pairs), and $p_i \in \{0, 1\}$ denote the ground-truth label.

# 5 EXPERIMENTS

## 5.1 EXPERIMENTAL SETTINGS

In this section, we first present the details of our experimental setup. We then conduct extensive experiments across multiple benchmark datasets, comparing the performance of our proposed method with several strong baselines. Finally, we provide an in-depth analysis of our model through ablation studies to investigate the contributions of each components.

**Datasets and Baselines.** In our experiments, we adopt five benchmark datasets from the Temporal Graph Benchmark (TGB) (Huang et al., 2023), namely tgbl-uci, tgbl-enron, tgbl-wiki, tgbl-subreddit, and tgbl-lastfm, which span a diverse range of domains. To comprehensively evaluate our proposed method, we compare it against seven popular dynamic graph learning methods: JODIE (Kumar et al., 2019), TGN (Rossi et al., 2020), TGAT (Xu et al., 2020), GraphMixer (Cong et al., 2023), TCL (Wang et al., 2021a), DyGFormer (Yu et al., 2023),FreeDyG (Tian et al., 2023) and RepeatMixer (Zou et al., 2024). Three representative time series models: BiLSTM (Hochreiter & Schmidhuber, 1997), iTransformer (Liu et al., 2024), and CATS (Kim et al., 2024), are also incorporated in our evaluation. The details of the datasets are provided in supplementary material.

Table 1: Comparison of link prediction performance between our proposed method and baselines in the transductive setting. Each experiment is repeated 5 times. Bold values indicate the best results.

| Category | Methods | tgbl-uci | | tgbl-enron | | tgbl-wiki | | tgbl-subreddit | | tgbl-lastfm | |
|---|---|---|---|---|---|---|---|---|---|---|---|
| | | MRR@20 | MRR@50 | MRR@20 | MRR@50 | MRR@20 | MRR@50 | MRR@20 | MRR@50 | MRR@20 | MRR@50 |
| Dynamic graph | JODIE | $63.45_{\pm 0.14}$ | $47.72_{\pm 0.38}$ | $43.78_{\pm 0.06}$ | $28.21_{\pm 0.20}$ | $80.84_{\pm 0.75}$ | $73.02_{\pm 1.22}$ | $88.25_{\pm 0.45}$ | $81.83_{\pm 0.62}$ | $40.81_{\pm 0.09}$ | $29.38_{\pm 0.07}$ |
| Dynamic graph | TGN | $63.97_{\pm 0.52}$ | $47.92_{\pm 0.64}$ | $27.02_{\pm 1.15}$ | $15.18_{\pm 1.71}$ | $88.13_{\pm 0.49}$ | $83.35_{\pm 0.73}$ | $89.57_{\pm 0.03}$ | $83.68_{\pm 0.11}$ | $46.11_{\pm 1.87}$ | $33.45_{\pm 1.97}$ |
| Dynamic graph | TGAT | $69.36_{\pm 0.38}$ | $52.88_{\pm 0.58}$ | $38.77_{\pm 0.08}$ | $23.23_{\pm 0.16}$ | $81.44_{\pm 0.61}$ | $74.77_{\pm 0.89}$ | $87.78_{\pm 0.07}$ | $81.34_{\pm 0.08}$ | $39.79_{\pm 0.17}$ | $30.57_{\pm 0.07}$ |
| Dynamic graph | GraphMixer | $72.40_{\pm 0.65}$ | $61.96_{\pm 0.05}$ | $53.14_{\pm 0.17}$ | $38.13_{\pm 0.14}$ | $83.46_{\pm 0.05}$ | $76.81_{\pm 0.34}$ | $86.80_{\pm 0.07}$ | $79.10_{\pm 0.10}$ | $45.70_{\pm 0.03}$ | $34.19_{\pm 0.24}$ |
| Dynamic graph | TCL | $63.20_{\pm 0.05}$ | $49.53_{\pm 0.03}$ | $40.43_{\pm 0.62}$ | $25.09_{\pm 0.49}$ | $86.48_{\pm 0.21}$ | $83.47_{\pm 0.28}$ | $87.49_{\pm 0.03}$ | $81.71_{\pm 0.06}$ | $48.45_{\pm 0.13}$ | $40.14_{\pm 0.14}$ |
| Dynamic graph | DyGFormer | $76.46_{\pm 0.04}$ | $69.89_{\pm 0.11}$ | $78.43_{\pm 0.22}$ | $69.90_{\pm 0.37}$ | $92.16_{\pm 0.11}$ | $89.95_{\pm 0.10}$ | $93.94_{\pm 0.03}$ | $91.06_{\pm 0.02}$ | $64.83_{\pm 0.01}$ | $55.01_{\pm 0.22}$ |
| Dynamic graph | FreeDyG | $80.46_{\pm 0.86}$ | $75.45_{\pm 0.97}$ | $77.86_{\pm 0.11}$ | $67.56_{\pm 0.81}$ | $93.56_{\pm 0.08}$ | $91.38_{\pm 0.02}$ | $93.64_{\pm 0.04}$ | $90.51_{\pm 0.01}$ | $64.16_{\pm 0.05}$ | $54.75_{\pm 0.06}$ |
| Dynamic graph | RepeatMixer | $79.82_{\pm 0.29}$ | $72.51_{\pm 0.23}$ | $79.42_{\pm 0.11}$ | $70.12_{\pm 0.19}$ | $92.84_{\pm 0.24}$ | $90.42_{\pm 0.52}$ | $\mathbf{94.51}_{\pm 0.12}$ | $\mathbf{92.17}_{\pm 0.15}$ | $70.66_{\pm 0.15}$ | $57.73_{\pm 0.12}$ |
| Time Series | BiLSTM | $70.85_{\pm 1.14}$ | $66.91_{\pm 1.12}$ | $77.27_{\pm 1.19}$ | $65.50_{\pm 1.26}$ | $88.63_{\pm 0.32}$ | $87.02_{\pm 0.53}$ | $83.36_{\pm 1.06}$ | $80.63_{\pm 1.12}$ | $70.10_{\pm 1.04}$ | $61.20_{\pm 1.25}$ |
| Time Series | iTransformer | $78.09_{\pm 0.12}$ | $71.80_{\pm 0.11}$ | $75.49_{\pm 0.14}$ | $64.00_{\pm 0.40}$ | $91.33_{\pm 0.28}$ | $88.66_{\pm 0.13}$ | $89.07_{\pm 0.10}$ | $84.07_{\pm 0.24}$ | $72.03_{\pm 0.20}$ | $61.98_{\pm 0.25}$ |
| Time Series | CATS | $72.06_{\pm 0.17}$ | $65.18_{\pm 0.23}$ | $79.84_{\pm 0.61}$ | $66.96_{\pm 0.77}$ | $90.21_{\pm 1.15}$ | $87.38_{\pm 0.68}$ | $78.58_{\pm 1.52}$ | $75.81_{\pm 1.56}$ | $72.91_{\pm 0.67}$ | $61.83_{\pm 0.36}$ |
| Joint | TSDyG (Ours) | $\mathbf{80.53}_{\pm 0.04}$ | $\mathbf{75.70}_{\pm 0.39}$ | $\mathbf{81.58}_{\pm 0.13}$ | $\mathbf{74.03}_{\pm 0.30}$ | $\mathbf{99.07}_{\pm 0.05}$ | $\mathbf{98.33}_{\pm 0.17}$ | $93.60_{\pm 0.05}$ | $91.02_{\pm 0.04}$ | $\mathbf{76.93}_{\pm 0.75}$ | $\mathbf{70.11}_{\pm 0.85}$ |

Table 2: Comparison of forecasting performance between our proposed method and baselines. Each experiment is repeated 5 times. Bold values indicate the best results.

| Category | Methods | tgbl-uci | | tgbl-enron | | tgbl-wiki | | tgbl-subreddit | | tgbl-lastfm | |
|---|---|---|---|---|---|---|---|---|---|---|---|
| | | MAE | MSE | MAE | MSE | MAE | MSE | MAE | MSE | MAE | MSE |
| Time Series | CATS | $0.338_{\pm 0.001}$ | $0.173_{\pm 0.001}$ | $0.341_{\pm 0.002}$ | $0.164_{\pm 0.001}$ | $0.135_{\pm 0.001}$ | $0.079_{\pm 0.001}$ | $0.482_{\pm 0.006}$ | $0.246_{\pm 0.001}$ | $0.391_{\pm 0.006}$ | $0.201_{\pm 0.002}$ |
| Joint | TSDyG | $\mathbf{0.267}_{\pm 0.002}$ | $\mathbf{0.136}_{\pm 0.001}$ | $\mathbf{0.242}_{\pm 0.003}$ | $\mathbf{0.125}_{\pm 0.004}$ | $\mathbf{0.131}_{\pm 0.002}$ | $\mathbf{0.071}_{\pm 0.002}$ | $\mathbf{0.121}_{\pm 0.002}$ | $\mathbf{0.060}_{\pm 0.001}$ | $\mathbf{0.308}_{\pm 0.009}$ | $\mathbf{0.160}_{\pm 0.003}$ |

**Evaluation Task and Metics.** In our experiments, we primarily focus on the future link prediction task, which is consistent with prior works. This task aims to predict the probability of an interaction occurring between two given nodes at a specific timestamp. It can be evaluated under two settings: the transductive setting, where all nodes are observed during training, and the inductive setting, where some nodes are unseen during training. We follow the dataset splits provided by TGB, dividing each benchmark into training, validation, and testing sets.

Following the Temporal Graph Benchmark (TGB), we formulate link prediction as a ranking problem by sampling multiple negative examples for each positive interaction. For a positive example $(u, v, t)$, we fix the source node $u$ and the timestamp $t$, and sample multiple negative destination nodes $\tilde{v}$. These negative nodes are either randomly selected or chosen from nodes that have interacted with $u$ but not at the current timestamp $t$. We adopt the Mean Reciprocal Rank (MRR) as the evaluation metric, as suggested in TGB. MRR is calculated as the reciprocal of the rank of the true destination node among all candidate (true and negative) destination nodes.

**Model Configurations.** In our model, the dimension of the codebook vectors $d_A$, time embeddings $d_T$ and projected embeddings $d_C$ is set to 172, 100 and 86, respectively. The hidden dimension $d_H$ is set to 172. The Cross-attention module consists of three layers, each with four attention heads. For all the baselines, we follow their official implementation settings to ensure a fair comparison.

**Implementation Details.** To adapt time series methods for our link prediction evaluation task, we first apply the proposed time series formulation module to convert the dynamic graph into time series representations compatible with the input requirements of these methods. The embeddings produced by the time series models are then used to predict the probability of interaction between node pairs. Unlike traditional regression tasks, we use binary cross-entropy loss instead of mean squared error during training. We employ the Adam (Kingma & Ba, 2015) optimizer with a learning rate of 0.0001 and adopt an early stopping strategy with a patience of 20 epochs, selecting the model that performs best on the validation set for final evaluation. In our experiments, we sample 20 and 50 negative examples per positive example, respectively. Each task is repeated five times, and all experiments are conducted on an NVIDIA RTX A40 GPU.

## 5.2 MAIN RESULTS AND DISCUSSIONS

Table 1 presents the performance of our method and baselines on transductive benchmarks. DyG-Former and FreeDyG outperform other models, highlighting their ability to capture temporal dependencies. However, their performance drops on larger graphs due to the high computational cost of sequential models like Transformers, which limits their scalability for long-range dependencies.

The results also suggest that time series methods can be effectively adapted to the link prediction task in dynamic graphs. By converting dynamic graphs into time series using our proposed time series formulation module, these methods can achieve competitive performance. To further validate

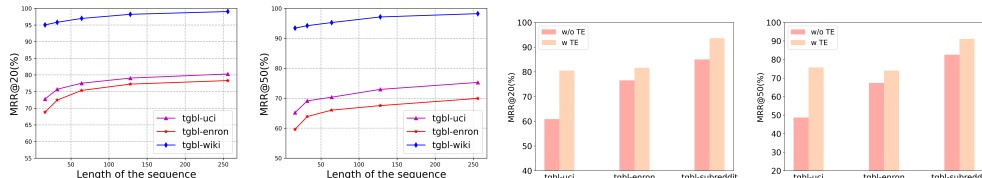

Figure 4: (a-b) Link prediction performance of our proposed TSDyG model with varying sequence lengths. (c-d) Link prediction performance of our proposed TSDyG model with and without the time embedding (TE).

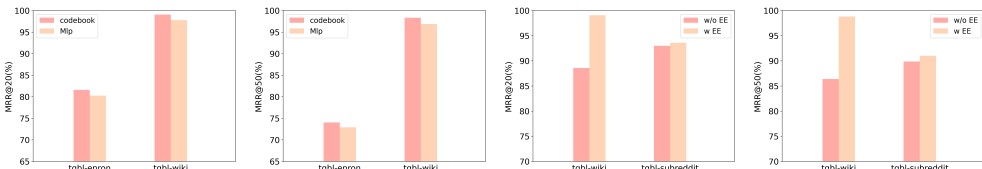

Figure 5: (a-b) Comparison of link prediction performance of our proposed TSDyG model with the codebook versus Mlp. (c-d) Link prediction performance of our proposed TSDyG model with and without the edge embedding (EE).

the effectiveness of time series-based approaches on binary data, we evaluate the performance of a traditional time series method (i.e., CATS (Kim et al., 2024)) and our proposed TSDyG model in a short-term forecasting setting. The fitness of each method on the binary time series data is assessed using Mean Squared Error (MSE) and Mean Absolute Error (MAE) between the output probability and the ground-truth value (either 0 or 1). The results, presented in Table 2, suggest that traditional time series methods are indeed applicable to the short-term forecasting of our formulated binary time series data. However, the performance of time series methods is not consistently strong across all datasets. This variability can be attributed to two main reasons. First, traditional time series models are primarily designed for multivariate, continuous time series data, whereas the time series derived from dynamic graphs in our formulation are binary and discrete. As a result, the design of these methods may not be well-suited for our setting. Second, most time series methods are tailored for forecasting tasks (i.e., predicting future values in a continuous sequence), whereas our task involves predicting the probability of interaction between specific node pairs at a given timestamp. This task discrepancy limits the direct applicability and effectiveness of standard time series models in the dynamic graph setting.

As shown in the results, our proposed TSDyG consistently outperforms the baselines across most tasks. The strong performance of TSDyG can be attributed to several key factors. Unlike previous dynamic graph methods that aggregate temporal dependencies from all neighboring nodes, our approach reformulates historical interactions between specific node pairs into binary time series. This targeted formulation allows the model to focus exclusively on the relevant interaction patterns. Moreover, TSDyG is capable of extracting temporal dependencies, including recurring interaction patterns, from both positive and negative interactions. This enhances the model's ability to distinguish true interactions from historically negative samples. Additionally, the integration of a cross-attention mechanism enables our model to capture long-range temporal dependencies more efficiently than traditional Transformer-based models. This advantage becomes especially prominent in large-scale dynamic graphs, contributing to the superior performance of our model.

Although our method approaches dynamic graphs through the lens of time series analysis, the proposed framework is better suited to modeling dynamic graphs than traditional time series methods. Conventional time series models are typically designed for multivariate, continuous-valued sequences, whereas our model introduces a discrete codebook tailored to handle the binary time series derived from dynamic graphs. This design enables more accurate representation of node dynamics. Furthermore, unlike standard time series approaches, our model explicitly leverages unique characteristics of dynamic graphs, such as edge features and temporal information. These additional modalities, often overlooked by traditional time series methods, enrich the representation learning process and contribute to the superior performance of our model on dynamic graph tasks.

## 5.3 Ablation Study

In the ablation study, we first examine the impact of sequence length on the performance of our proposed model. We vary the sequence length across 16, 32, 64, 128, and 256, and conduct experiments on tgbl-uci, tgbl-enron, and tgbl-wiki. The results, presented in Figure 4, indicate that performance generally improves with longer sequence lengths. This is because longer sequences allow the model to capture more comprehensive temporal patterns from historical interactions, leading to more accurate future predictions.

Next, we examine the contribution of the time embedding to the overall performance of our model. We fix the sequence length and compare model performance with and without the time embedding. Experiments are conducted on tgbl-uci, tgbl-enron, and tgbl-subreddit. The results, shown in Figure 4, indicate a significant performance drop when the time embedding is removed. This highlights the importance of time embedding, which encodes the temporal context of interactions. It enables the model to more accurately capture the behavioral patterns of nodes. Without the temporal signal, the model struggles to distinguish between positive interactions and historical (negative) examples, resulting in reduced performance.

We also evaluate the effectiveness of the codebook component in our model by replacing it with a standard Mlp. Experiments are conducted on tgbl-enron and tgbl-wiki. As shown in Figure 5, replacing the codebook with an MLP results in a slight performance degradation, suggesting that the codebook is a more effective choice for modeling temporal dependencies in dynamic graphs.

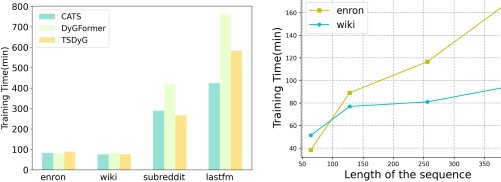

Figure 6: (a) The training time of our proposed TSDyG and baselines across benchmark datasets. (b) The training time of our proposed TSDyG with different sequence lengths.

Finally, we evaluate the impact of edge features on the performance of our proposed model in attributed dynamic graphs. Experiments are conducted on tgbl-wiki and tgbl-subreddit. The results, shown in Figure 5, reveal a noticeable performance drop when edge features are removed. These findings highlight the importance of edge features in enhancing the expressiveness of node embeddings and enable the model to more effectively distinguish temporal dependencies across different node pairs.

## 5.4 Running Time Analysis

To evaluate the efficiency of our proposed model, we measure the training time of TSDyG model and compare it against baseline models, including CATS (Kim et al., 2024) and DyGFormer (Yu et al., 2023). For a fair comparison, the sequence length is set to 128 for both CATS and TSDyG, while DyGFormer uses a maximum sequence length of 48 due to its architectural constraints. The evaluation is conducted across multiple benchmark datasets, and the results are presented in Figure 6. These results show that our model is more computationally efficient than traditional CTDG methods. In addition, we assess the impact of sequence length on the computational cost of TSDyG model using the tgbl-enron and tgbl-wiki datasets. As illustrated in Figure 6, training time increases as the sequence length grows, which is expected due to the higher computational demand associated with processing longer temporal contexts.

## 6 Conclusion

We review prior work on discrete-time and continuous-time dynamic graph learning, highlighting limitations such as loss of fine-grained temporal information, difficulty modeling long-range dependencies, and neglect of non-existing interactions. To address these, we propose transforming interactions into time series and introduce the TSDyG model. Experiments show that TSDyG effectively captures temporal dependencies and achieves strong performance on multiple benchmarks. However, our model may under-perform in high-surprise dynamic graphs with low edge repetition. Addressing such scenarios remains an important direction for future work.

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

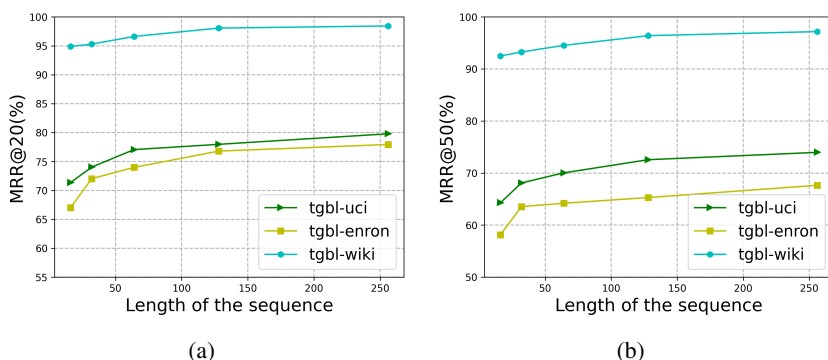

Figure 7: Link prediction performance of our proposed TSDyG model with varying sequence lengths in the inductive setting. The evaluation is performed on tgbl-uci, tgbl-enron and tgbl-wiki.

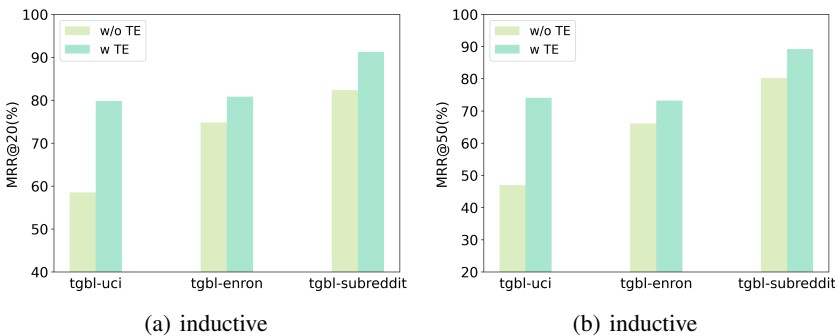

Figure 8: Link prediction performance of our proposed TSDyG model with and without the time embedding (TE).

## A  APPENDIX

### A.1  EXPERIMENTAL SETTINGS

The details of the Temporal Graph Benchmark (TGB) Huang et al. (2023) are summarized in Table 4. Among these datasets, tgbl-wiki and tgbl-subreddit include attributed edge features, while the others do not. For our TSDyG model and the time series baselines, we use a sequence length of 512 for tgbl-enron and tgbl-subreddit, and 256 for the remaining datasets. For dynamic graph baselines, we follow the model configurations specified in their original papers. All models are trained for 60 epochs, and the best checkpoint is adopted for evaluation.

### A.2  MAIN RESULTS

Table 3 presents the link prediction performance of our proposed TSDyG model compared to dynamic graph and time series baselines in the inductive setting across five datasets from the Temporal Graph Benchmark. Notably, TSDyG shows substantial gains on datasets like tgbl-enron, tgbl-wiki and tgbl-lastfm, highlighting its strength in capturing long-range temporal dependencies and recurring patterns. These findings demonstrate the effectiveness of our approach in enhancing temporal modeling of dynamic graphs in the inductive setting.

### A.3  ABLATION STUDY

We investigate the impact of sequence length on our model's performance in the inductive setting by varying it from 16 to 256 on tgbl-uci, tgbl-enron, and tgbl-wiki. As shown in Figure 7, performance

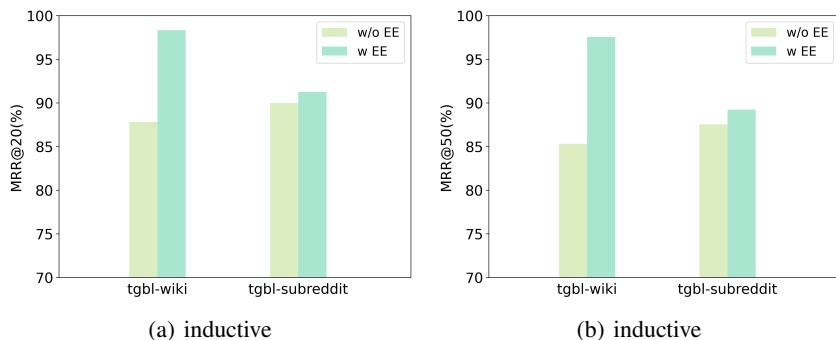

Figure 9: Link prediction performance of our proposed TSDyG model with the codebook and Mlp.

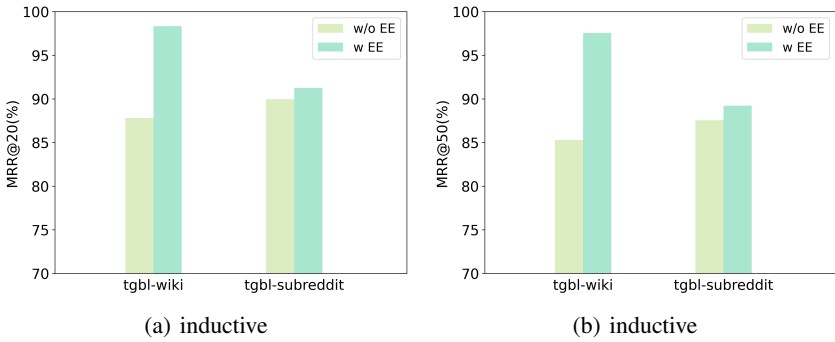

Figure 10: Link prediction performance of our proposed TSDyG model with and without the edge embedding (EE).

consistently improves with longer sequences, mirroring the trend observed in the transductive setting. This highlights the importance of capturing longer temporal histories for accurate prediction.

We further assess the impact of time embedding on model performance in the inductive setting by comparing results with and without it, using a fixed sequence length. Experiments on tgbl-uci, tgbl-enron, and tgbl-subreddit (Figure 8) reveal a noticeable performance drop when the time embedding is removed, particularly on tgbl-uci. This underscores the critical role of time embedding in capturing the temporal context of interactions across both inductive and transductive settings.

We also investigate the effectiveness of the codebook component in our model by replacing it with a standard Mlp in the inductive setting. Experiments on tgbl-enron and tgbl-wiki (Figure 9) show a slight performance drop when using the MLP, indicating that the codebook is better suited for modeling binary time series data in our approach.

Finally, we evaluate the impact of edge features on our model's performance in attributed dynamic graphs under the inductive setting. Experiments on tgbl-wiki and tgbl-subreddit (Figure 10) show a clear performance drop when edge features are removed. This underscores the importance of edge features in enriching node representations and helping the model better capture temporal dependencies across diverse node pairs.

### A.4 LIMITATION

Our proposed model relies on the historical recurring interactions of node pairs to capture temporal dependencies. However, its effectiveness may be limited in dynamic graphs with the low repeat ratio, as such settings do not provide sufficient information from individual node pair interactions alone. In these cases, incorporating information from neighboring nodes becomes necessary to better model the temporal dynamics. We leave this direction for future work.

Table 3: Comparison of link prediction performance between our proposed method and baselines in the inductive setting. Each experiment is repeated 5 times. Bold values indicate the best results.

| Category | Methods | tgbl-uci | | tgbl-enron | | tgbl-wiki | | tgbl-subreddit | | tgbl-lastfm | |
|---|---|---|---|---|---|---|---|---|---|---|---|
| | | MRR@20 | MRR@50 | MRR@20 | MRR@50 | MRR@20 | MRR@50 | MRR@20 | MRR@50 | MRR@20 | MRR@50 |
| Dynamic graph | JODIE | $58.83_{\pm 0.13}$ | $42.44_{\pm 0.22}$ | $36.45_{\pm 0.41}$ | $25.66_{\pm 0.30}$ | $75.45_{\pm 0.93}$ | $63.99_{\pm 1.52}$ | $82.35_{\pm 1.31}$ | $78.12_{\pm 1.20}$ | $36.58_{\pm 1.46}$ | $25.89_{\pm 1.58}$ |
| Dynamic graph | TGN | $59.37_{\pm 0.81}$ | $42.36_{\pm 1.07}$ | $25.05_{\pm 1.02}$ | $13.55_{\pm 1.25}$ | $87.88_{\pm 0.34}$ | $82.70_{\pm 0.32}$ | $81.68_{\pm 0.21}$ | $83.68_{\pm 0.11}$ | $44.66_{\pm 1.14}$ | $31.23_{\pm 1.44}$ |
| Dynamic graph | TGAT | $65.20_{\pm 1.20}$ | $49.95_{\pm 1.64}$ | $30.85_{\pm 0.26}$ | $12.82_{\pm 0.31}$ | $80.06_{\pm 0.64}$ | $73.78_{\pm 0.49}$ | $84.68_{\pm 1.32}$ | $75.47_{\pm 1.77}$ | $38.19_{\pm 0.45}$ | $29.03_{\pm 0.09}$ |
| Dynamic graph | GraphMixer | $71.38_{\pm 0.35}$ | $59.37_{\pm 0.52}$ | $51.12_{\pm 0.18}$ | $29.05_{\pm 0.09}$ | $81.17_{\pm 0.05}$ | $74.89_{\pm 0.18}$ | $83.08_{\pm 0.09}$ | $76.95_{\pm 0.27}$ | $43.59_{\pm 0.07}$ | $32.27_{\pm 0.18}$ |
| Dynamic graph | TCL | $57.90_{\pm 0.44}$ | $44.35_{\pm 0.02}$ | $36.48_{\pm 0.50}$ | $20.03_{\pm 0.17}$ | $85.55_{\pm 0.08}$ | $82.17_{\pm 0.23}$ | $86.25_{\pm 1.59}$ | $78.23_{\pm 1.13}$ | $47.02_{\pm 0.44}$ | $38.33_{\pm 0.54}$ |
| Dynamic graph | DyGFormer | $75.20_{\pm 0.06}$ | $68.53_{\pm 0.17}$ | $75.96_{\pm 0.09}$ | $67.84_{\pm 0.20}$ | $90.96_{\pm 0.07}$ | $87.29_{\pm 0.17}$ | $91.10_{\pm 0.30}$ | $89.71_{\pm 0.10}$ | $63.90_{\pm 0.11}$ | $53.49_{\pm 0.29}$ |
| Dynamic graph | FreeDyG | $77.19_{\pm 0.11}$ | $68.56_{\pm 0.74}$ | $61.83_{\pm 1.22}$ | $52.62_{\pm 2.81}$ | $92.96_{\pm 0.23}$ | $89.57_{\pm 0.46}$ | $83.95_{\pm 2.65}$ | $78.85_{\pm 2.43}$ | $61.80_{\pm 0.60}$ | $50.13_{\pm 0.75}$ |
| Dynamic graph | RepeatMixer | $78.80_{\pm 0.37}$ | $71.60_{\pm 0.02}$ | $71.23_{\pm 0.51}$ | $68.89_{\pm 0.35}$ | $92.33_{\pm 0.27}$ | $89.93_{\pm 0.16}$ | $\mathbf{92.00}_{\pm 0.18}$ | $\mathbf{90.74}_{\pm 0.51}$ | $68.49_{\pm 0.18}$ | $55.88_{\pm 0.09}$ |
| Time Series | BiLSTM | $67.90_{\pm 0.98}$ | $64.97_{\pm 0.90}$ | $76.35_{\pm 1.02}$ | $63.92_{\pm 1.24}$ | $86.76_{\pm 0.21}$ | $84.92_{\pm 0.26}$ | $80.38_{\pm 1.24}$ | $77.45_{\pm 1.08}$ | $68.37_{\pm 0.85}$ | $59.42_{\pm 0.64}$ |
| Time Series | iTransformer | $76.91_{\pm 0.08}$ | $70.92_{\pm 0.08}$ | $74.84_{\pm 1.07}$ | $63.54_{\pm 0.11}$ | $90.98_{\pm 0.53}$ | $87.61_{\pm 0.38}$ | $88.51_{\pm 0.05}$ | $83.58_{\pm 0.11}$ | $70.39_{\pm 0.28}$ | $60.28_{\pm 0.33}$ |
| Time Series | CATS, | $70.18_{\pm 0.24}$ | $63.16_{\pm 0.27}$ | $78.07_{\pm 0.56}$ | $65.48_{\pm 0.64}$ | $88.11_{\pm 1.21}$ | $84.41_{\pm 1.22}$ | $77.67_{\pm 0.11}$ | $73.70_{\pm 0.25}$ | $71.88_{\pm 0.37}$ | $60.81_{\pm 0.91}$ |
| Joint | TSDyG (Ours) | $\mathbf{79.83}_{\pm 0.08}$ | $\mathbf{74.08}_{\pm 0.13}$ | $\mathbf{80.83}_{\pm 0.22}$ | $\mathbf{73.23}_{\pm 0.82}$ | $\mathbf{98.80}_{\pm 0.06}$ | $\mathbf{97.55}_{\pm 0.02}$ | $91.26_{\pm 0.02}$ | $89.22_{\pm 0.04}$ | $\mathbf{75.64}_{\pm 0.05}$ | $\mathbf{68.95}_{\pm 0.43}$ |

Table 4: The details of benchmarks.

| Dataset | Domain | #Nodes | #Edges | #Steps | Attributed |
|---|---|---|---|---|---|
| tgbl-uci | social | 3,212 | 59,835 | 58,911 | No |
| tgbl-enron | social | 365 | 125,235 | 22,632 | No |
| tgbl-wiki | rating | 9,227 | 157,474 | 152,757 | Yes |
| tgbl-subreddit | rating | 10,984 | 672,447 | 588,915 | Yes |
| tgbl-lastfm | recommendation | 1,980 | 1,293,103 | 1,283,614 | No |

## A.5 USE OF LARGE LANGUAGE MODELS

In our work, we solely employ LLMs to check grammatical errors and refine sentence structures.

