# OpenReview forum: "Revisiting Dynamic Graphs from the Perspective of Time Series"
_ICLR.cc/2026/Conference — ICLR 2026 Conference Withdrawn Submission_

### Official Review · Reviewer_Az8W · 2025-10-17

**Soundness:** 2
**Presentation:** 2
**Contribution:** 1
**Rating:** 2
**Confidence:** 4

**Summary:**

This work offers a reframing of dynamic graphs as pairwise binary time series, supported by a lightweight architecture that leverages cross-attention for long-range modeling.

**Strengths:**

1. The paper highlights the underexplored signal from non-existing interactions and represents pairwise interactions as binary time series is somewhat intuitive.

**Weaknesses:**

1. The authors claim to model pairwise links as binary time series, where each node pair’s interaction history is encoded as a sequence of [0,1] values and then combined with time and edge encodings before being processed by a cross-attention layer. However, this design appears conceptually close to the neighbor co-occurrence encoding used in DyGFormer, which also represents node-pair histories but through interaction counts rather than binary indicators. The authors should clarify the substantive distinction between their binary formulation and count-based co-occurrence encoding, and provide a concrete justification for its advantages. Specifically, they should explain whether the binary time series captures additional temporal information (e.g., gaps, recency, or interval structure) that the count-based encoding cannot, and demonstrate through controlled experiments that this representation yields consistent improvements rather than being a renaming of the existing co-occurrence approach.

2. The experiments do not align with established protocols in the literature (e.g., DyGFormer). Only a single negative sampling strategy is adopted, whereas recent works typically report results under random，historical，inductive negative sampling settings. Moreover, several standard continuous-time dynamic graph benchmarks—such as MOOC, Social Evo, Flights, UNtrade, and Contact—are omitted, limiting the comparability of the results. In addition, the evaluation relies solely on MRR, which provides an incomplete view of model performance. Common metrics such as AUROC, Average Precision and F1 should also be reported to assess both ranking quality and discriminative capability more comprehensively.

3. Since the authors frame their model as a time series formulation, the time encoding component should play a crucial role. However, the paper only adopts the time encoding scheme from GraphMixer and includes an ablation that simply removes it. This provides little insight, as time encoding has already been shown to be essential across dynamic graph models. A more informative analysis  should be explored how different time encoding strategies affect performance.

4. The training time comparison should include all baseline models to ensure a fair and meaningful evaluation. Reporting training efficiency for only two models provides an incomplete picture and may bias the assessment of scalability.


5. The overall presentation is weak, particularly in the Method section, which lacks clarity and organization. Key components and notations should be defined more systematically to improve readability and technical transparency.

**Questions:**

Please see the weaknesses above.

---

### Official Review · Reviewer_NRw2 · 2025-10-26

**Soundness:** 2
**Presentation:** 2
**Contribution:** 2
**Rating:** 2
**Confidence:** 4

**Summary:**

This paper proposes a new relative encoding for the transformer-based dynamic graph neural network, where for each target edge $(u,v,t)$, its element for a historical interaction $(u,x,t')$ is 1 if $ x=v$ and $0$ otherwise. Besides this, the authors also modify the self-attention into cross-attention when converting the historical edge sequence into a node representation, reducing the computational complexity.

**Strengths:**

The proposed method is more lightweight than previous transformer-based methods, but can achieve a similar performance on the selected datasets.

**Weaknesses:**

- The novelty of this paper is limited. The authors claim that their method “approaches dynamic graph learning from a time series perspective.” However, the overall pipeline closely resembles that of DyGFormer. In essence, the main contributions are modifying the relative encoding in DyGFormer to a binary version and replacing self-attention with cross-attention, which are relatively minor changes.
- The experimental results presented in this paper are not fully convincing. First, with the exception of tgbl-wiki, the selected TGB datasets are small-scale and rarely used, which limits the relevance of the results to broader community practices. It is recommended that the authors report results on the datasets from the tgbl-leaderboard (i.e., tgbl-wiki, tgbl-review, tgbl-coin, tgbl-comment, tgbl-flight) to enhance comparability and credibility. Second, the reviewer is concerned about whether the hyperparameters of the baseline methods were properly tuned for a fair comparison, particularly for those baselines applied to new datasets.

**Questions:**

Please see the Weaknesses.

---

### Official Review · Reviewer_Um3i · 2025-10-31

**Soundness:** 1
**Presentation:** 2
**Contribution:** 2
**Rating:** 2
**Confidence:** 4

**Summary:**

The present work considers link prediction in continuous-time dynamic graphs. The authors argue that times when no interactions occur between two nodes are informative and point out that existing TGNNs typically ignore this data. To address this shortcoming, the authors propose converting CTDG to a binary time series: for every node pair, they construct a binary sequence with one entry per timestamp, indicating whether there is an interaction between the two nodes at that timestamp. Naturally, these timeseries contain the same information as the original CTDG, but the authors use it somewhat differently: during training, when they consider a specific node pair, they derive a fixed-length time series for each of the two interacting nodes based on when they interact, including with other nodes, which leads to 0-entries in the interactions between the two nodes of interest, thus incorporating information about when the nodes do not interact with each other. The authors propose a model, which they call TSDyG, and which contains a time-series formulation module, an embedding-generation module, and a cross-attention module. TSDyG is designed to leverage the information encoded in the time-series representation of the CTDG by learning representations as a concatenation of embeddings created via a "discrete codebook", an encoding of timestamps, and an encoding of edges. The cross-attention module then learns relevant information for time-aware node embeddings. The authors evaluate their approach on five benchmark networks from TGB, where it somewhat outperforms the selected baselines.

I find the core idea of the paper quite interesting and insightful, that is, the idea that non-interactions between two nodes (u,v) at times t when u or v interact with other nodes carry useful information for learning interaction patterns between u and v. However, I must say that this core idea is somewhat hidden in the details and could have been presented more clearly. I also find that, in general, while the authors use the appropriate references, use reasonable baseline models, and present the technical details of their approach, the paper lacks some storytelling that would make it easier for the reader to follow along and understand the ideas. Moreover, it seems to me like the authors make some statements that are not warranted based on the experiments they have conducted (see weaknesses below). In addition, there are several (quite many) minor issues that need attention (see below, under questions).

**Strengths:**

- The authors present an insightful idea of considering times of non-interactions between node pairs of interest as an additional source of information to learn their interaction patterns for dynamic link prediction.
- The authors propose a TGNN that performs competitively with the selected baselines in an empirical evaluation on five benchmark datasets from TGB.

**Weaknesses:**

- The authors make a claim that I believe is unsubstantiated by their results. Specifically, they claim that "the performance of time series methods is not consistently strong across all datasets", however, they have only tested one "time series method" on only five datasets, which cannot be sufficient to make such a general claim.
- The claim that TSDyG is "more computationally efficient than traditional CDTG methods" is also not supported by enough evidence. The authors show empirical wall-clock training times, but compare their method only against one TGNN for CDTG graphs and one "traditional" time-series model (CATS) on four datasets, which cannot be sufficient to make such a general claim either.
- While the authors mention limitations, they only do so briefly as a side note in the conclusion.

**Questions:**

There are a couple of things that remained unclear to me, and I hope the authors can help me clarify those points.

1. I am a little bit puzzled by Figure 1. What is its intended message, and where does the time-series (0,1,1,0) come from?
2. You state that your "design facilitates the modeling of long-range temporal dependencies from historical data while maintaining lower computational complexity compared to traditional self-attention mechanisms.". However, it is not clear to me what exactly about your design enables better computational efficiency. Could you elaborate?
3. In your contributions, you state that "TSDyG effectively captures recurring interaction patterns between nodes and models long-term temporal dependencies in dynamic graphs.". How can we know that TSDyG actually captures recurring interaction? For example, do the selected empirical datasets exhibit large amounts of such recurring interactions?
4. Could your suggested approach for generating time series essentially be understood as a novel negative link sampling approach? I get this idea because the time-series conversion from a CTDG essentially contains the same amount of information as the CTDG -- unless the CTDG is directed and not all links are reciprocated; in that case, some information is lost. But otherwise, the CTDG can be recovered from the time-series representation. However, the way the 0 entries in the time series entries per node pair are generated reminds me of negative samples, and in this case, it seems like they are chosen in a quite useful/informative way. What do you think about this perspective?
5. I am still not sure what the "discrete codebook" is and how it works. The description in l.259 says that it contains two entries, and l.262 says that those two entries are vectors which represent the interactions between the node pair. This suggests to me that there is a codebook for every node pair, that is, a total of $n^2$ codebooks. Is that correct? And can you provide some intuition regarding the idea behind the codebook and how it can be interpreted?
6. Under "Evaluation Task and Metics", you mention that this work focuses *primarily* on future link prediction. Was there an intention to include other tasks? Or is there another task included, and I have just missed it?
7. When you mention that you used "their official implementation settings" for all methods, do you mean the settings from each original paper? Or some settings used for entries on the TGB leaderboard? Or yet something different?
8. You mention that DyGFormer and FreeDyG outperform other models, but that "their performance drops on larger graphs due to the high computational cost of sequential models like Transformers, which limits their scalability for long-range dependencies.". To be honest, I don't follow the argument. A high computational cost should not lead to worse performance on larger graphs, rather, it should lead to longer computation times. But if the models were adjusted for larger graphs, for example, by using fewer embedding dimensions or attention heads, then I would agree that the performance might drop. What is it you mean here?
9. I am wondering what failure modes TSDyG has. You mention that it may underperform in high-surprise dynamic graphs with low edge repetition. What about, for example, scenarios with missing data, which is quite common in empirical networks? How would missing/incomplete data affect TSDyG?

Minor points
- L.58 "time-sensitive prediction task" sounds like the prediction task has some time-critical aspects, such as in real-time computing, which I do not believe is the case here. Perhaps this should be rephrased.
- I suppose that, in section 3, it should say "~The~ A discrete-time dynamic graph", and "~The~ A continuous-time dynamic graph". Also, the formulation "usually" suggests that there are other representations of continuous-time dynamic graphs that may be of interest here.
- I suppose you mean "node pair" in l.189 instead of "pair node"?
- In l.194, do you mean $\mathbb{R}^{d_E}$ instead of $\mathbb{R}^d_E$? Also, do you mean "dimensionality of the edge feature" instead of "dimension of the edge feature"?
- I believe the "the" in l.197 should not be there: "representation learning on ~the~ time series-based dynamic graphs".
- The word "capturing" in l.198 doesn't seem to fit.
- I also believe that the "the" in l.199 should not be there: "is evaluated through ~the~ link prediction".
- I think I know what you mean by "important interactions information" in l.212, however, you may want to revise the sentence for clarity.
- L.244-245 mentions "a binary time series sequence", which sounds like a list of time series, however, I believe that only one time series is meant?
- I think it should say "The" instead of "In" in l.258: "~In~ The embedding generation module comprises...".
- I believe a "the" is missing in l.283, and perhaps it should be "to" instead of "on": "We apply a linear transformation ~on~ *to the* joint embedding..."
- "Inspired by ~the~ previous work" in l.288 is somewhat unspecific and should be supported by references and perhaps some words to briefly describe the relevant previous work.
- I am not sure what the reader learns from the sentence "This design allows the model to distil the most relevant temporal information." Hopefully, all models aim to "distil the most relevant information".
- L.318 mentions that the selected TGB datasets "span a diverse range of domains", however, it is not mentioned what those domains are.
- Typo in l.342: "Metics" should be "Metrics".

---

### Official Review · Reviewer_j6DG · 2025-11-08

**Soundness:** 2
**Presentation:** 1
**Contribution:** 2
**Rating:** 2
**Confidence:** 4

**Summary:**

The authors of this work propose to address temporal graph learning from the perspective of binary time series modelling, where each edge defines a time series and the presence or absence of edges at a given time stamp is encoded as a time series of binary numbers. They then apply a time encoder to the resulting multi-dimensional time series, and use a cross-attention model to capture temporal patterns. The approach is evaluated in a link prediction task for five data sets, showing performance improvements for four of the five data sets. Apart from the link prediction setting, the authors also address a forecasting scenario and perform a study how different aspects and parameters of the approach influence the results.

**Strengths:**

[S1] The paper addresses an interesting and current problem in deep learning for continuous-time dynamic graphs

[S2] The paper is generally well written and easy to follow (notwithstanding the required clarifications on the methods and experimental evaluation mentioned below).

**Weaknesses:**

- [W1] The article takes a rather limited perspective on related work on temporal graph analysis and learning, which is restricted to works from a narrow subset of the deep graph learning community, missing related approaches from network science, where temporal networks have been a focus of study for more than a decade. See my comments under Q1.

- [W2] Related to the previous point, I do not agree that the idea to apply methods for time series learning/analysis to address temporal graph learning tasks like link prediction is a novelty of this work, see Q2 below.

- [W3] The approach to model temporal graphs as binary time series, where edges that do not occur at a given timestamp are explicitly represented by zeros, is not well-motivated and it is not clear how the edge time series are actually defined, see Q4.

- [W4] Even though the authors specifically address continuous-time temporal graphs, they seem to model the temporal graph as a (discrete) sequence of binary values, indicating the presence or absence of an edge at certain discrete times, which seems like a bad fit. See Q5.

- [W5] There are several open questions regarding the experimental evaluation. This includes the use of the TGB benchmark data, the negative sampling strategy used for the link prediction task, the use of batch-based splits, the difference between the prediction and forecasting setting, the choice of hyperparameters, the ablation study, as well as several other aspects, see Q6 - Q9 below.

**Questions:**

- [Q1] I do not think that the coverage of related works in continuous time temporal graph learning is representative of what has been done (especially considering works outside a small circle of temporal graph learning researchers). In particular, the authors limit their review of related works on memory- and model-centric approaches, missing out a host of works (both in deep learning and network science) that have considered, e.g. time-respecting path structures in continuous-time temporal graphs, which fall in neither category, and which capture patterns that will likely be missed by the authors' approach.

- [Q2] Related to the previous point, the authors argue that the time series perspective on temporal graphs is a novelty of this work, however there are other uncited works that take a similar perspective, e.g. modelling temporal graphs as time series of adjacency matrices (which is in a way very similar to what the present paper does). See, e.g.

- Faith W. Mutinda et al.: Time Series Link Prediction Using NMF, DEIM Forum, 2019
- P da Silva Soares, R Prudencio: Time Series Based Link Prediction, 2012
- E Richard et al.: Link Prediction in Graphs with Autoregressive Features, 2014
- FS Passino et al. Link prediction in dynamic networks using random dot product graphs, 2019

I would argue that those works take a similar perspective and should at least be cited (or possibly even included as baselines). In any case, it will be important to discuss this work to highlight the research gap addressed by the current work. Do the authors disagree?

- [Q3] I am not convinced by the argument that existing temporal graph learning methods do not capture the absence of edges. I think this boils down to the difference of an edge list vs. adjacency matrix representation, where edge lists focus on existing edges (implicitly assuming that edges not in the list are absent) while adjacency matrices use a full square-sized representation of a graph. Why is the authors' approach superior to other works that represent temporal graphs as sequences of adjacency matrices?

- [Q4] Considering that this work specifically address continuous-time temporal graphs, i.e. sequences of edge events, it seems strange to represent such temporal graphs as binary time series. This requires an implicit discretization, since we need to represent the absence of edges at specific points in time. If I understand correctly, the authors propose that the time points at which the absence of an edge is represented in the sequence are implicitly given by the timetamps of other events in the temporal graph.

One issue that I had with this work is that it is not clear to me at which timepoints we sample the "absence of edges". In the example in Fig. 2 the time series for edge u-a includes t3 (in which neither node u nor node a aree active, but it does not include t2. The formal definition does not clarify this as well.

Can the authors clearly explain how the edge time series are defined? Why not, e.g., sample at evenly spaced time points (which would make this more similar to discrete-time graphs)?

Also, could the authors motivate what additional information the inclusion of absent edges provides over a list of existing edges (which implies that other edges are absent)?

- [Q5] In the motivation, the authors state that other approaches to model discrete-time temporal graphs that have represented temporal graphs as sequences of graphs lead to "redundant and computationally expensive graph sequences". But is this not what the authors do implicitly as well by considering edge-level time series that include zeros at intermediate timestamps, especially when the aggregate graph is dense and edge activities follow a heavy-tail distribution?

- [Q6] Although the TGB benchmark is often used in the temporal graph learning community, I am not convinced that it is the best approach to rigorously evaluate the proposed method. This may be a matter of personal taste, but I believe that TGB has several issues that make it a bad choice to evaluate e.g. link prediction, e.g. the use of batch- rather than time-based splits in data with very different temporal activity distributions, the fact that naive methods (like Edgebank) have been shown to outperform sophisticated architectures in some of the data sets, and the rather limited selection of domains for the data. I understand that the authors may disagree this with, and given that TGB is still used in the community, I do not consider this criticism in my evaluation of the paper. However, I still believe that the paper would benefit from using data from more domains (e.g. using temporal graphs from the netzschleuder repository) and from using time-based rather than batch-based training and evaluation.

- [Q7] In section 5.2 the authors write that they use a negative sampling strategy where they "sample multiple negative destination nodes v. These negative nodes are either randomly selected or chosen from nodes that have interacted with u but not at the current timestamp t."

What does this "either or" mean. The authors do not seem to reports for both strategies. Please clarify this.

Similarly, in section 5.2 the authors discuss the impact of the sequence length on the performance, but they do not mention how this parameter has been chosen in the main results in section 5.1.

- [Q8] From the very short description in section 5.2 I could not follow how the forecasting setup has been implemented and evaluated and how it differs from the prediction setup. I would expect that this uses a time- rather a batch-based split of the data. Moreover, the authors mention a "short-term" forecasting setting but do not really explain the details. From the reference to the loss function, one could guess that only a single next edge or absence of edge is predicted, but this would cause problems in case multiple events are observed at the same time stamp. Please explain.

In general, the current description of the experimental evaluation (and some parts of the methods) is insufficient and this must be clarified before the paper can be published.

- [Q9] This may be nitpicking, but I would argue that most of the experiment in the section "ablation study" are actually not ablations. An ablation study selectively ablates (i.e. removes) components from a proposed model to study how different components impact performance. However, here the authors study the impact of the sequence length, which is a hyperparameter, they replace the "binary codebook" by an MLP (which I would argue adds a component rather than removing one) and they study the role of edge features. The only part that I would see as an ablation study is the experiment that removes the time embedding. I would suggest to change the wording.

---

### Note · Authors · 2025-11-16

I have read and agree with the venue's withdrawal policy on behalf of myself and my co-authors.